# The Cdc14 Phosphatase Controls Resolution of Recombination Intermediates and Crossover Formation during Meiosis

**DOI:** 10.3390/ijms22189811

**Published:** 2021-09-10

**Authors:** Paula Alonso-Ramos, David Álvarez-Melo, Katerina Strouhalova, Carolina Pascual-Silva, George B. Garside, Meret Arter, Teresa Bermejo, Rokas Grigaitis, Rahel Wettstein, Marta Fernández-Díaz, Joao Matos, Marco Geymonat, Pedro A. San-Segundo, Jesús A. Carballo

**Affiliations:** 1Center for Biological Research Margarita Salas, Department of Cellular and Molecular Biology, Spanish National Research Council (CSIC), Ramiro de Maeztu 9, 28040 Madrid, Spain; paula.alonso@cib.csic.es (P.A.-R.); davalvar@ucm.es (D.Á.-M.); katkastr@outlook.com (K.S.); pasilcar@gmail.com (C.P.-S.); carmente.96@gmail.com (T.B.); marta.fernandez@cib.csic.es (M.F.-D.); 2Department of Cell Biology, Charles University, Viničná 7, 12843 Prague, Czech Republic; 3Genome Damage and Stability Centre, University of Sussex, Brighton BN1 4DY, UK; george.garside@gmail.com; 4Leibniz Institute for Age Research/Fritz Lipmann Institute (FLI), Beutenbergstr. 11, D-07745 Jena, Germany; 5Institute of Biochemistry, HPM D6.5-ETH Zürich, Otto-Stern-Weg 3, 8093 Zürich, Switzerland; arterm@mskcc.org (M.A.); rokas.grigaitis@univie.ac.at (R.G.); rahel.wettstein@univie.ac.at (R.W.); joao.matos@univie.ac.at (J.M.); 6Memorial Sloan Kettering Cancer Center, New York, NY 10065, USA; 7Max Perutz Labs, University of Vienna, Dr. Bohr-Gasse 9, 1030 Vienna, Austria; 8Department of Genetics, University of Cambridge, Downing Street, Cambridge CB2 3EH, UK; m.geymonat@gen.cam.ac.uk; 9Institute of Functional Biology and Genomics (IBFG), Spanish National Research Council (CSIC) and University of Salamanca, 37007 Salamanca, Spain; pedro.ss@csic.es

**Keywords:** Cdc14, Yen1, Sgs1, Mus81, CDK1, Ndt80, Cdc5, Cdc20, meiotic recombination, aneuploidy, Holliday junction

## Abstract

Meiotic defects derived from incorrect DNA repair during gametogenesis can lead to mutations, aneuploidies and infertility. The coordinated resolution of meiotic recombination intermediates is required for crossover formation, ultimately necessary for the accurate completion of both rounds of chromosome segregation. Numerous master kinases orchestrate the correct assembly and activity of the repair machinery. Although much less is known, the reversal of phosphorylation events in meiosis must also be key to coordinate the timing and functionality of repair enzymes. Cdc14 is a crucial phosphatase required for the dephosphorylation of multiple CDK1 targets in many eukaryotes. Mutations that inactivate this phosphatase lead to meiotic failure, but until now it was unknown if Cdc14 plays a direct role in meiotic recombination. Here, we show that the elimination of Cdc14 leads to severe defects in the processing and resolution of recombination intermediates, causing a drastic depletion in crossovers when other repair pathways are compromised. We also show that Cdc14 is required for the correct activity and localization of the Holliday Junction resolvase Yen1/GEN1. We reveal that Cdc14 regulates Yen1 activity from meiosis I onwards, and this function is essential for crossover resolution in the absence of other repair pathways. We also demonstrate that Cdc14 and Yen1 are required to safeguard sister chromatid segregation during the second meiotic division, a late action that is independent of the earlier role in crossover formation. Thus, this work uncovers previously undescribed functions of the evolutionary conserved Cdc14 phosphatase in the regulation of meiotic recombination.

## 1. Introduction

Meiotic recombination is initiated by the conserved Spo11 transesterase, which introduces numerous DNA Double-Strand Breaks (DSBs) into the genome [1]. The association of single-strand DNA binding proteins, including RPA, Rad51 and the meiosis specific recombinase, Dmc1, with the resected DSB ends promotes strand invasion into the intact homologous non-sister chromatid template, which culminates in the formation of a displacement loop (D-loop). Nascent D-loops can be processed through Synthesis-Dependent Strand Annealing (SDSA) repair pathway to generate non-crossover (NCO) repair products [2]. Alternatively, the stabilization of the D-loop followed by second-end capture gives rise to even more stable structures, such as double Holliday Junction (dHJ) intermediates. In budding yeast meiosis, dHJs are most frequently resolved into crossovers (COs) through the combined action of Mlh1-Mlh3 (MutLγ) and Exo1 [3,4,5]. A second class of COs arises from the resolution of recombination intermediates via the Structure-Selective Endonucleases (SSEs) Mus81-Mms4, Slx1-Slx4 and Yen1 [6,7,8,9,10]. Finally, Sgs1, the ortholog of the Bloom’s syndrome helicase (BLM), can also promote dissolution of JMs via the Sgs1-Top3-Rmi1 (STR) complex, which would exclusively generate NCOs products. In addition, multiple rounds of strand exchange over a stabilized D-loop, can give rise to three and four interconnected duplexes, also known as multichromatid joint molecules (mc-JMs) [11]. Further, both DSB-ends might also participate in multi-invasion events on two or three chromatids, creating heavily branched DNA structures [12]. These complex DNA species must be efficiently processed to prevent the risk of becoming potential hazards for the genome integrity due to their capacity to form aberrant crossovers, or other forms of toxic repair products. Correct processing of all types of JMs requires the orchestrated action of a set of helicases, topoisomerases and endonucleases, capable of dislodging multi-branched DNA assemblies. The BLM/Sgs1, in addition to be involved in SDSA repair, plays a prominent role in eliminating aberrant “off-pathway” JMs, including mc-JMs [7,8,13,14,15,16]. 

In budding yeast, it is not fully clear which enzymatic complexes take over the function of removing any unprocessed branched DNA intermediates outside prophase I, due to the existence of numerous SSEs capable of processing such substrates [17,18]. Upon the initiation of *NDT80*-dependent transcription, and the activation of the budding yeast polo-like kinase Cdc5, cells abandon the pachytene stage concomitantly with a surge of enzymatic activity that culminates in the resolution of ZMM-dependent and ZMM-independent dHJs to give rise to COs and some NCO products [19,20]. Cdc5 phosphorylation is required to stimulate the function of a set of SSEs, most notoriously Mus81-Mms4 [9,21]. The Mus81-Mms4 complex acts on branched DNA substrates that have not been resolved by the canonical ZMM-dependent Class-I CO pathway [4,13]. On the other hand, Yen1^GEN1^, a conserved member of the Rad2/XPG family of SSEs, appears to act later in meiosis, despite the fact that Yen1 is considered a prototypical HJ resolvase [22]. In meiotic cells, this discrepancy is easily explained since the activity of Yen1 is negatively regulated by CDK1-dependent phosphorylation, preventing the activation of the nuclease during prophase I [9]. Elimination of CDK-phosphorylation sites in Yen1 prematurely turns on its enzymatic activity and promotes its nuclear localization. Therefore, counteracting the phosphorylation of Yen1 generates the fully active form of the nuclease [9,10]. In mitotic cells, the dephosphorylation of Yen1 is carried out by the Cdc14 phosphatase [23,24]. The activation of Yen1 during anaphase by Cdc14 allows for the resolution of persistent repair intermediates that would otherwise impose a physical impediment to chromosome segregation. It is currently unknown whether Cdc14 also activates Yen1 in meiosis [4,9,25]. Furthermore, it is possible that Cdc14 dephosphorylates other substrates important for the timely resolution of different types of JMs [25].

Cdc14 is a well-conserved dual-specificity phosphatase that has been defined as a key component in the reversal of CDK phosphorylation during exit from mitosis [26]. In budding yeast, Cdc14 activity is essential and cells lacking this phosphatase remain arrested at the end of mitosis [27,28,29]. Additionally, Cdc14 regulates transcriptional silencing at the rDNA and other repeated sequences [30,31,32,33,34,35,36]. Several pathways regulate Cdc14 localization and activity. The RENT complex retains Cdc14 at the nucleolus until anaphase through its interaction with the Cif1/Net1 protein [37,38]. Cdc14 is released from the nucleolus by the sequential action of two pathways. The FEAR (Cdc Fourteen Early Anaphase Release) pathway promotes the early release of Cdc14 through the phosphorylation of Cif1/Net1 and Cdc14 [39,40,41,42,43,44,45]. Later in anaphase, the Mitotic Exit Network (MEN) keeps Cdc14 in its released state, allowing for the dephosphorylation of additional substrates, which is important for the full termination of mitosis and cytokinesis [46,47].

Notably, a number of functions have been allocated to Cdc14 before its anaphase-triggered release. In particular, Cdc14 has been involved in the completion of the replication of late-replicating regions, such as the rDNA locus and other parts of the genome [48]. Additionally, the anaphase-independent release of Cdc14 has been observed upon induction of DNA damage; the generation of DSBs promotes the transitory release of Cdc14 from the nucleolus, targeting the SPB component, Spc110, for dephosphorylation [49].

Cdc14 is highly conserved and orthologs have been identified in other organisms, including fission yeast, nematodes, insects, and vertebrates [50]. The human genome contains three Cdc14 paralogs, hCDC14A, hCDC14B and hCDC14C [51,52]. The depletion of Cdc14A, which is required for mitotic CDK inactivation, leads to defects in centrosome duplication, mitosis and cytokinesis [53]. In vertebrates, CDC14B exits the nucleolus in response to DNA damage [54,55], a process conserved in fission yeast [56], and more recently also observed in budding yeast [49]. Thus, in addition to playing important roles in the DNA Damage Response (DDR) [54] it appears that both Cdc14A and Cdc14B may be required for the efficient repair of damaged DNA [55]. 

Cdc14 is also critically required for the completion of meiosis [57]. In budding yeast, Cdc14 is released from the nucleolus in two waves; the first during anaphase I and the second during anaphase II [58,59,60]. Contrary to mitotic cells, Cdc14 release from the nucleolus requires the essential function of the FEAR pathway, but MEN appears to be dispensable [61,62,63]. Additionally, some components of the MEN pathway, such as Cdc15, have functionally differentiated to fulfil a role in spore morphogenesis [61,62]. Curiously, the inactivation of the FEAR pathway in meiosis allows for exclusively one round of nuclear division to take place, culminating in the formation of asci carrying two diploid spores (dyads) [58,59,61,64,65]. The premature activation of FEAR blocks spindle assembly during meiosis, a process normally averted by PP2A^Cdc55^ [42,66,67,68]. On the other hand, the inactivation of Cdc14 function by means of FEAR mutants, or by employing conditional temperature-sensitive alleles of *CDC14*, impairs the second round of chromosome segregation [58,59]. These interesting observations prompted the identification of another critical function of Cdc14 at the meiosis I to meiosis II transition, which is the licensing of SPB re-duplication/half-bridge separation by the asymmetrical enrichment of Cdc14 on a single SPB during anaphase I [69]. Furthermore, the absence of the phosphatase creates chromosome entanglements, a problem that can be reverted by deleting *SPO11* [59]. Surprisingly, albeit it was known that removing meiotic recombination could improve chromosome segregation in the *cdc14* mutant, nothing was known about how the recombination machinery and Cdc14 crosstalk during meiosis.

Here, we describe a previously undefined role of Cdc14 in budding yeast meiosis. We reveal that Cdc14 is involved in ensuring correct meiotic recombination by gradually implementing Yen1 activity following Cdc5 activation. We show that, at anaphase I, Yen1 already shows a prominent nuclear localization promoted by Cdc14. Yen1 reaches its maximum activity during meiosis II, and Cdc14 is strictly required for such activation. A constitutively active form of Yen1, termed Yen1^ON^, which is insensitive to CDK downregulation, compensates for Cdc14 deficiency, suggesting that Cdc14 normally reverses CDK phosphorylation in Yen1 during meiosis. Unexpectedly, Cdc14 also promotes Yen1 activity in *CDC20*-depleted cells, allowing for JM processing and CO formation when other repair pathways are absent. Thus, Yen1 plays a more prominent role during meiotic recombination than originally anticipated, which can be manifested even before meiosis I division is normally completed. Notably, Yen1 complements other endonucleases whose activities are mainly constrained to the time-window between the end of prophase I and before anaphase I. We propose that Yen1, controlled by Cdc14, implements its resolvase function when other enzymatic activities begin to decline outside of their optimum operative time-window by the end of prophase I, and that Yen1 covers a wider gap extending its action until the end of the second meiotic division.

## 2. Results

### 2.1. Identification of a Separation-Of-Function Allele of CDC14 Defective in Sporulation

The reversal of CDK1 phosphorylation is carried out in budding yeast by the conserved Cdc14 phosphatase. This essential protein can only be studied utilizing conditional *cdc14* alleles that fully or partially impair its functionality. Thermosensitive mutants have been widely used in the past for mitotic and meiotic studies. However, meiotic recombination is sensitive to variations of temperature [70], thus we considered using other types of conditional alleles to further examine the role of Cdc14 in meiosis. During the course of a different study, we generated the *cdc14-HA* allele, carrying an endogenous C-terminal full-length tagged version of *CDC14* with three copies of the hemagglutinin epitope (HA). Unlike other temperature-sensitive alleles of *cdc14* used in many studies (see introduction), mitotic growth was not affected in diploid cells carrying the *cdc14-HA* allele incubated at different temperatures (Figure 1A). Unexpectedly, the *cdc14-HA* strain was highly defective for sporulation, displaying only ~10% of asci, containing one or two spores (Figure 1B,C), and lacking dityrosine autofluorescence, an indicator of spore wall maturation [71] (Figure 1D,G and Appendix A). Thus, we conclude that *cdc14-HA* cells cannot efficiently complete sporulation despite their apparently normal mitotic growth and may be a useful tool to explore additional aspects of Cdc14 meiotic function.

### 2.2. Distinct Sporulation-Defective Alleles of CDC14 Block Meiotic Progression at Different Stages

To determine the cause(s) of the sporulation defect of the *cdc14-HA* mutant we analyzed the progression of the meiotic program by studying the kinetics of meiotic DNA replication and nuclear divisions. During the mitotic cell cycle, released Cdc14 dephosphorylates replication factors, such as Sld2 and Dpb2 [72], and the lack of Cdc14 activity leads to problems during termination of DNA replication [48]. However, FACS analysis of DNA content revealed that S-phase progression occurred with nearly identical kinetics in both control and *cdc14-HA* strains (Figure 1E), suggesting that the meiotic S-phase is not altered in *cdc14-HA*. Next, meiotic divisions were monitored in synchronous cultures of *CDC14* and *cdc14-HA*. We found that *cdc14-HA* cells displayed slightly slower and less efficient kinetics of nuclear divisions; nevertheless, most *cdc14-HA* cells exited the one nuclei stage, although with a ~60 min delay compared to wild-type cells (Figure 1F and Appendix A). Previous work has shown that most meiotic cells in *cdc14* temperature-sensitive mutants are blocked with two nuclei, due to the inability to assemble fully functional tetrapolar spindles at meiosis II [57,59,67,73]. However, we noticed that the *cdc14-HA* mutant progressed beyond the two-nuclei stage in a high proportion of cells (Appendix A), distinguishing this *cdc14-HA* allele from the temperature-sensitive mutants used in other studies.

Next, we generated an additional meiosis-specific allele of *CDC14* by replacing its endogenous promoter with the *CLB2* promoter, which is repressed to a great extent in meiotic cells [74]. In this case, a 3HA tag was located at the N-terminal region of the protein. Cells expressing *P_CLB2_-3HA-CDC14* (hereafter, *cdc14-md*, for meiotic depletion) recapitulated the sporulation defect of *cdc14-HA* (Figure 1C). The most striking difference was that, although the first meiotic nuclear division took place in *cdc14-md*, as it occurs in *cdc14-HA*, most cells remained binucleated in *cdc14-md*, even at late time points in meiosis (Appendix A). Thus, the terminal meiotic phenotype of *cdc14-md* appears to be different from that of *cdc14-HA*.

### 2.3. Cdc14 Protein Levels Are Depleted during Meiosis in cdc14-HA Cells

We next used Western blot to analyze Cdc14 protein levels during meiosis in *CDC14*, *cdc14-md* and *cdc14-HA* strains. The *cdc14-HA* mutant displayed reduced levels of the tagged protein, even at time zero, prior to meiotic induction, and they were further reduced after 6–8 h into meiosis (Figure 2A and Appendix A). The depletion of Cdc14 was more effective in the *cdc14-md* allele, while in *cdc14-HA* it occurred more gradually (Figure 2A). This is not related to delayed entry into the meiotic program, since all strains underwent meiotic S-phase with similar kinetics, as determined by FACS analysis (Figure 2A). To directly ascertain if Cdc14 protein levels varied from mitotically dividing cells to meiotic cells, we also analyzed extracts from exponentially growing cultures compared to extracts prepared from cultures immediately after meiotic induction (Figure 2B). The wild-type strain showed similar amounts of Cdc14 protein in both mitotic and meiotic cultures. The levels of Cdc14-HA were lower in mitotic cells, compared to the wild type, and they were further reduced upon meiotic entry. On the other hand, Cdc14 protein levels in the mitotically dividing *cdc14-md* mutant were similar to the wild type, but they showed a drastic depletion with meiosis onset (Figure 2B). Increasing the copy number of different *cdc14* alleles, as well as the overproduction of the Cdc14-HA protein restored sporulation to a large extent (Figure 1G and Appendix A). These results indicate that the reduction in Cdc14 phosphatase levels, but not its full depletion, causes the meiotic phenotype in the hypomorphic *cdc14-HA* mutant. On the other hand, the almost complete absence of the protein throughout the entire meiotic process likely provokes the more stringent block in nuclear divisions observed in *cdc14-md* cells. Thus, both alleles are useful tools to investigate different meiotic events that could be regulated by Cdc14.

### 2.4. SPB Integrity Is Compromised in cdc14-HA Cells after the Second Meiotic Division

As mentioned above, a great proportion of *cdc14-HA* cells eventually passed the two-nuclei stage, observed as the terminal meiotic phenotype in *cdc14-md* (this study) and *cdc14* temperature-sensitive mutants [57,59,73]. Thus, we monitored the kinetics and morphology of the spindle pole bodies (SPBs) and spindles during the first and the second meiotic divisions (Figure 2C,D). We observed that, by eight hours into meiosis, *CDC14* and *cdc14-HA* cultures contained around 75% and 60% of cells with more than one SPB, respectively. Among the population of cells that showed multiple SPBs, 56% of wild-type cells presented more than two; therefore, they had initiated meiosis II. However, only 40% of *cdc14-HA* cells displayed more than two SPBs, confirming a slight reduction in the number of cells forming meiosis II spindles carrying four separated SPBs. This observation correlates with the slightly lower frequency of *cdc14-HA* cells containing more than two DAPI masses (Figure 2D and Appendix A). As expected, after 24 h of meiotic induction, the wild type showed over 90% of cells with three or four SPBs. In marked contrast, most *cdc14-HA* meiotic mutant cells lacked a single SPB, suggesting that, in the absence of Cdc14, the structural integrity of SPBs becomes compromised once both meiotic divisions take place (Figure 2D and Appendix A). On the other hand, most cells of the *cdc14-md* mutant, contained only 2 SPBs for the entire length of the time course (Appendix A), confirming earlier studies revealing a role for the phosphatase in SPB reduplication/half-bridge separation [69]. Additionally, the average time taken to complete both meiotic divisions was not significantly different in the *cdc14-HA* mutant, compared to the wild type (Appendix A). Thus, we can conclude that, albeit with some difficulty, the *cdc14-HA* mutant is still capable of undergoing both meiotic divisions, but it is unable to form spores. On the other hand, the *cdc14-md* mutation, which produces a more drastic meiosis-specific depletion of Cdc14, prevents the separation of duplicated SPBs after the first division; therefore, these cells are unable to assemble proper tetrapolar spindles at meiosis II. Such a behavior more closely resembles the phenotype(s) previously described for other non-meiosis specific *cdc14* mutants [59,69].

### 2.5. Impeded Homolog Separation and Sister Missegregation Are Common Hallmarks of cdc14 Mutations in Meiosis

We took advantage of the hypomorphic *cdc14-HA* allele, which allows us to study the second division in the absence, or with very little amount, of the Cdc14 protein to analyze chromosome segregation. In the *cdc14-HA* mutant, cells were frequently observed with more than four DAPI-stained masses late in meiosis (Appendix A). This might be due to impediments during chromosome segregation in the first and/or the second meiotic division. To test this possibility, we analyzed nuclear morphology, along with the meiotic spindle, in individual cells undergoing the first and second divisions (Figure 3A(i–v)). Cells presenting meiosis I (MI) spindles with a length of ≥5 µm were considered to be in late anaphase I and, therefore, two DAPI masses on each pole should be distinguishable (Figure 3A(i)). On the other hand, cells presenting DAPI-stained bridges might reflect some form of chromosome entanglement (Figure 3A(ii,iii); [59]). Wild-type cells showed a clear separation of nuclear content in 74% of anaphase I cells analyzed (Figure 3A). By contrast, in both *cdc14-HA* and *cdc14-md* mutants, only ~20% of anaphase I cells displayed two fully resolved nuclei (Figure 3A). A large proportion (~80%) of anaphase I mutant cells presented a chromosomal bridge, suggesting that, despite having undergone MI spindle elongation correctly, some form of DNA entwining was not properly resolved.

A subtle delay in cohesin removal during anaphase I onset was observed in a small proportion of *cdc14-HA* cells (Figure 3B,C and Appendix A), which may be responsible for at least a fraction of these bridged DAPI masses [59]. To test if these entanglements were eventually eliminated or persisted until the second meiotic division, we examined metaphase II cells and scored the configuration of their nuclei (Figure 3A(iv,v)). The wild type presented over 76% of metaphase II cells with single individualized nuclei. By contrast, 75% of metaphase II cells in the *cdc14-HA* mutant showed DNA threads connecting both nuclear masses. As mentioned above, the severity of the SPB re-duplication/half-bridge separation phenotype in the *cdc14-md* strain precluded us to identify metaphase II cells with 4-SPBs in *cdc14-md* cells. We conclude that diminished levels of Cdc14 in meiosis results in the abnormal formation of nuclear bridges after the first meiotic division and that these bridges persist at least until metaphase II.

To determine whether the DNA bridges present at metaphase II in *cdc14-HA* mutant cells were due to problems arising during chromosome segregation, we constructed strains carrying fluorescent markers on specific chromosomes and followed their segregation [75]. We made use of *tetO* repeats integrated at the interstitial *lys2* locus of chromosome II, together with the presence of the *tetR* repressor gene fused with GFP, which associates to these repeats ([76,77]; Appendix A). First, we analyzed strains labelled at chromosome II on both homologs to determine chromosome segregation fidelity in cells that had completed both meiotic divisions. We found that chromosome II segregated correctly in 95% and 78% of wild type and *cdc14-HA* tetranucleated cells, respectively. The remaining 22% of *cdc14-HA* mutant cells displayed at least one of the four nuclei, lacking a GFP focus (Figure 3D and Appendix A type 1, 2 and 3 only). We further analyzed the first and the second division separately using strains carrying lacO repeats at the trp1 locus, which is linked to the centromere of chromosome IV (*CENIV*) and expressing the *lacI* repressor fused to GFP [78] (Appendix A). In order to assess homolog disjunction, we first examined binucleated cells homozygous for the *CENIV-GFP*, and we did not find any defect in the segregation of homologs during the first meiotic division in *cdc14-HA* (Figure 3E). Very similar results were previously reported for homozygous *CENV-GFP* markers in *cdc14-1* mutants [67]. Next, we analyzed sister chromatid segregation in the second meiotic division using heterozygous strains for the *CENIV-GFP* marker. In this case, 50% of tetranucleated *cdc14-HA* cells displayed problems segregating sister centromeres (Figure 3F). Taking all of these results together, we conclude that accurate sister chromatid segregation in meiosis requires unperturbed Cdc14 levels.

### 2.6. Preventing Recombination Alleviates Defective Events in Meiosis-Specific cdc14 Mutants

It has been shown that the elimination of meiotic DSBs alleviates problems with chromosome segregation in FEAR and *cdc14-1* mutants [59]. Therefore, we considered that the presence of anaphase I nuclear bridges and sister chromatid missegregation in *cdc14-HA* cells could be originated, at least in part, from problems caused by defective recombination. If this were the case, these problems should be alleviated by eliminating meiotic recombination. To test this hypothesis, we combined the *cdc14-HA* allele with the *spo11-Y135F* mutation, which prevents DSB formation [1]. The absence of DSBs improved the separation of DAPI masses in anaphase I in both *cdc14-HA* and *cdc14-md* mutants (Figure 3A), indicating that recombination leads to the formation of anaphase I bridges in both meiosis-defective *cdc14* mutant strains. Accordingly, the inactivation of Spo11 catalytic activity also improved sporulation in *cdc14-HA* and *cdc14-md* (Figure 3G). Thus, these results suggest that, during meiosis, Cdc14 plays a role in promoting the accurate repair of Spo11-dependent DSBs.

### 2.7. Meiotic Recombination Is Impaired in cdc14 Mutants

To gain insight into the function of Cdc14 during meiotic recombination, we performed DNA physical assays using the well-characterized *HIS4LEU2* recombination hotspot (Figure 4A,B; [70,79,80,81]) in wild type, *cdc14-HA* and *cdc14-md* strains. First, we measured total levels of DSBs, CO and NCOs products. Levels of DSBs and COs were very similar in the three strains analyzed, albeit a small, not significant, reduction in NCO levels was observed in *cdc14-HA* and *cdc14-md* mutants (Figure 4C,D). Although a strong effect in the formation of recombination products was not perceived in *cdc14* mutant cells, the shorter, or longer, lifespan of JMs could alter the ratio of NCO to CO to compensate for lower, or higher, efficiency in the processing of these intermediates [11,82,83]. To address this possibility, we analyzed JM formation in the *ndt80Δ* mutant background [84]. Accumulation of JMs takes place in *ndt80Δ* cells due to the lack of *CDC5*-dependent JM-resolution promoting activity [20]. DSBs, NCOs and JMs formed efficiently in the *cdc14-HA* mutant (Figure 4E,F). Thus, under unchallenged conditions, Cdc14 appears to have little or no effect on the initiation and processing of meiotic recombination during prophase I, at least at the *HIS4LEU2* hotspot. 

Next, we wanted to monitor if CO and/or NCO formation was affected in *cdc14* mutants when DSB repair is moderately compromised by eliminating one of the two main resolution pathways described in budding yeast, such as the Mus81-Mms4 pathway [6]. Thus, we proceeded to evaluate the efficiency of JM resolution in a *mus81Δ ndt80Δ* mutant background. In order to obtain an accurate reading of repair product formation we first confirmed that total levels of JMs were similar in both *mus81Δ ndt80Δ* and *mus81Δ cdc14-HA ndt80Δ* mutant combinations (Figure 4B and Appendix A). To promote JM resolution in the absence of Ndt80, we took advantage of the widely employed *CDC5*-inducible expression system (*CDC5-IN*), where the expression of the *CDC5*, which is required for the resolution of JMs into COs, can be induced at any point required in the meiotic time-course by the addition of β-estradiol (ES) to the media [20,85]. We added ES after 7 h in meiosis and monitored CO and NCO formation. The induced expression of *CDC5* in the *mus81Δ* mutant efficiently led to the formation of COs only one hour after ES addition to the medium (Figure 5A,B); CO levels did not increase further after the 8 h time-point. On the other hand, NCOs steadily increased, reaching their maximum at 24 h. In contrast, fewer COs were detected in the *mus81Δ cdc14-HA* mutant. NCOs levels were also lower in *mus81Δ cdc14-HA*, and total levels did not rise over the period when *CDC5* was expressed (Figure 5B). Thus, Cdc14 is required for the efficient formation of COs and NCOs, at least when meiotic DSB repair takes place in the absence of the Mus81-Mms4-dependent resolution pathway. 

To confirm that Cdc14 plays a relevant role in the repair of recombination intermediates, at least when certain resolution pathways are compromised, we decided to study the efficiency of repair in a *NDT80* background using 2D-gels that provide more precise information of the different DNA species involved during the JM resolution process (Figure 4B). To this end, we took advantage of the well-known negative effect on DSB-repair when depleting Sgs1 in combination with mutations in one or more of the SSEs, such as *mms4/mus81* and *yen1/slx1/slx4* [4,13]. In this case, we used the *cdc14-md* allele, to prevent any residual activity of the Cdc14 phosphatase. As expected, the elimination of *YEN1* in the *sgs1-md mms4Δ* background led to the increased accumulation of unprocessed JMs compared to the *sgs1-md mms4Δ* double mutant (Figure 5C,D). Strikingly, after 24 h in meiosis, the *sgs1-md mms4Δ cdc14-md* triple mutant also accumulated great levels of unresolved JMs compared to the *sgs1-md mms4Δ* double mutant (Figure 5C,D).

Interestingly, the accumulation of unresolved JMs in *sgs1-md mms4Δ cdc14-md* was similar to that of the *sgs1-md mms4Δ yen1Δ* triple mutant (Figure 5C,D). Furthermore, CO levels at the *HIS4LEU2* hotspot were highly reduced in the *sgs1-md mms4Δ cdc14-md* strain; a phenotype that was nearly identical to that of *sgs1-md mms4Δ yen1Δ* (Figure 5E,F). Thus, in the absence of the Sgs1 and Mus81-Mms4-dependent pathways for processing recombination intermediates, the lack of the Cdc14 phosphatase leads to the same defects as with the absence of the Yen1 resolvase, strongly suggesting that Cdc14 and Yen1 are involved in the same repair pathway during meiosis.

### 2.8. Yen1 Nuclear Localization and Activity during Meiosis Requires Cdc14 Function

In mitotic cells, Yen1 localization is cytoplasmic when CDK activity is high, but release of Cdc14 during anaphase I reverts CDK-dependent phosphorylation on Yen1, promoting its nuclear entry and activity [9,23,24,86]. In meiotic cells, Yen1 subcellular localization is also regulated in a phosphorylation-dependent and cell-cycle stage-specific manner [9]. To determine whether Cdc14 also regulates Yen1 subcellular distribution during meiosis, we examined nuclear localization of Yen1 in anaphase I cells, and we found that it was reduced in *cdc14-HA* compared to the wild type (Figure 6A). We also tested if the enzymatic activity of Yen1 is controlled by Cdc14 during meiosis. Samples were collected from synchronized meiotic cultures of wild-type and *cdc14-HA* strains carrying a functional 9MYC-tagged version of *YEN1* [10]. From each time point taken, Yen1 was immunoprecipitated from the meiotic extracts and the beads were incubated with a synthetic HJ substrate [16]. Active Yen1 should be able to cleave the HJ substrate, whereas the phosphorylated inactive form of the nuclease would not [24]. In wild-type cells, Yen1 did not display significant nuclease activity during early stages of the meiotic time course prior to the accumulation of Cdc5 (Figure 6B), but approximately 2 h after Cdc5 induction, we detected robust HJ processing, consistent with the activation of Yen1 during meiotic divisions (Figure 6B,C; [9]). Notably, the resolvase activity displayed by Yen1 in the *cdc14-HA* mutant was extremely low for the duration of the whole time course (Figure 6B,C). The meiotic S-phase was completed with similar timing in wild-type and *cdc14-HA* cells, suggesting that the initiation of the meiotic program occurred with similar kinetics in both strains (Figure 6D). Moreover, the Cdc5 protein (marking the exit from prophase I) was detected at the same time, and at the same levels, in both wild-type and *cdc14-HA* strains (Figure 6B). Thus, these results strongly suggest that Cdc14 promotes the nuclear accumulation of Yen1 during anaphase I and is required for its enzymatic activity at times when Cdc5 activity is high.

### 2.9. Constitutively Active Yen1 Suppresses the Accumulation of Aberrant Recombination Intermediates in cdc14 Mutants

To determine whether the activation of Yen1 by Cdc14 is exerted by controlling its phosphorylation status, we next used a phosphorylation-resistant version of the nuclease (*YEN1^ON^*) in which the mutation of nine of the CDK consensus sites of phosphorylation in Yen1 renders the protein constitutively active [10,24]. We combined the *cdc14-HA* allele with *YEN1^ON^* and examined sporulation efficiency. Similar to the elimination of Spo11-dependent DSBs, the unrestrained activity of Yen1 partly rescued the sporulation defect of *cdc14-HA* (Figure 6E). We also directly tested the effect of *YEN1^ON^* in chromosome segregation during both meiotic divisions (Figure 6F). First, we checked if the presence of Yen1^ON^ promotes the resolution of the DNA bridges observed at late anaphase I in *cdc14-HA* mutants. Indeed, a two-fold improvement in *cdc14-HA YEN1^ON^* over the *cdc14-HA* single mutant was observed (Figure 6F, left graph). Furthermore, the mis-segregation of sister chromatids was also fully rescued to the levels of the *YEN1^ON^* single mutant in the *cdc14-HA YEN1^ON^* double mutant (Figure 6F, right graph). Moreover, to confirm if the improvement in chromosome segregation by *YEN1^ON^* was due to the re-establishment of Yen1 resolvase activity, we employed two types of assays. First, we checked using 2D-gels whether the persistent accumulation of JMs and/or CO formation observed at the *HIS4LEU2* hotspot in the repair-deficient *sgs1-md mms4Δ cdc14-md* triple mutant was alleviated by the constitutively active Yen1^ON^. Notably, the introduction of *YEN1^ON^* efficiently prevented the accumulation of unresolved JMs after 24 h in meiosis (Figure 5C,D). Furthermore, JMs were efficiently resolved giving rise to high levels of COs in the *sgs1-md mms4Δ cdc14-md YEN1^ON^* quadruple mutant (Figure 5E,F). Finally, we used the in vitro resolution assay to determine whether Yen1^ON^ was capable of restoring resolvase activity in the *cdc14-HA* mutant. We found that, unlike Yen1, the constitutively active Yen1^ON^ displayed potent resolvase activity in the absence of Cdc14 (Figure 6G). Altogether, these results indicate that Yen1 activation by Cdc14 is important for the repair of unprocessed recombination intermediates in meiosis.

### 2.10. Cdc14 and Yen1 Promote JM Resolution during the First Meiotic Division

In the meiotic program, Yen1 activity appears to reach its maximum level during the second meiotic division [9,10]. Nevertheless, it is possible that, during the first release of Cdc14 at meiosis I, Yen1 can be activated to act over remnants of unprocessed or complex JMs. In support of this idea, we observed that 52% of wild-type cells displayed nuclear Yen1 in anaphase I (Figure 6A), indicating that a high proportion of Yen1 molecules have been translocated to the nucleus during the first meiotic division. To test if Cdc14 is acting over Yen1 already in meiosis I or, by contrast, Yen1 is exclusively activated during the second release of Cdc14 at meiosis II, we used the *cdc20-md* mutant to deplete the APC component Cdc20 and arrest cells in meiosis I [85]. We analyzed the effect of Cdc14 and Yen1 on JM resolution in *cdc20-md* cells blocked at the first meiotic division when the Sgs1 and Mus81-Mms4 repair pathways, normally acting at the end of prophase I, have been depleted. Remarkably, the *cdc20-md sgs1-md mms4Δ yen1Δ* and *cdc20-md sgs1-md mms4Δ cdc14-md* quadruple mutants exhibited much higher levels of unprocessed JMs than the *cdc20-md sgs1-md mms4Δ* triple mutant (Figure 7A,B). In addition, this JM resolution defect was accompanied by a markedly reduced capacity of CO formation in both *cdc20-md sgs1-md mms4Δ yen1Δ* and *cdc20-md sgs1-md mms4Δ cdc14-md* quadruple mutants compared to the *cdc20-md sgs1-md mms4Δ* triple mutant (Figure 7C,D). Together, we conclude that, when the action of other relevant JM processing pathways is compromised, Cdc14 and Yen1 are required for proper JM resolution as early as in the first meiotic division.

## 3. Discussion

Meiotic recombination is fundamental for sexual reproduction to ensure that homologous chromosomes are accurately distributed to the gametes, as well as to facilitate distinct allele combinations that sustain evolution. Homologous recombination is initiated by the introduction of programmed DSBs followed by resection, homology search, and DNA strand exchange. Those early steps in the recombination process lead to the formation of stable JMs, which are ultimately resolved into two main classes of HR repair products, known as COs and NCOs (Figure 8). Recombining chromosomes may also contain intermediates consisting of three- and four-armed DNA structures, such as mc-JMs, where three and four DNA duplexes stay connected [7,8,81,87,88,89,90,91]. In some cases, unresolved recombination intermediates can persist until the metaphase to anaphase transition, where a subset of late-acting nucleases takes charge of their processing in order to safeguard genome integrity [17]. The unrestrained activity of such nucleases can interfere with the correct allotment of COs and NCOs in meiosis [10]; thus, they are tightly controlled by several layers of regulation (Figure 8). Phosphorylation events carried out by key cell cycle kinases, such as CDK, DDK, and Polo Kinase, are the most extended form of regulation of these enzymatic activities [9,21]; therefore, it is expected that phosphatases can play also crucial roles in this regulation. In *S. cerevisiae* mitotic cells, at least one of the late acting nucleases, Yen1, is modulated by the highly conserved Cdc14 phosphatase [23,24,86]. Although a number of studies have unveiled the relevance of Cdc14 during meiotic chromosome segregation and SPB/centrosome dynamics, a role for Cdc14 in the regulation of meiotic recombination has not yet been established. Given the essentiality of *CDC14* in the budding yeast, most previous meiotic studies have used conditional temperature-sensitive alleles of *cdc14* that are blocked in meiosis after the first round of chromosome segregation [27,59,69,73]. Despite the large amount of valuable data that has been collected over the years using these *ts* alleles, other less conspicuous functions of Cdc14, for example those affecting meiosis II, might have been precluded from being discovered. Here, using two different meiosis-specific alleles of *CDC14* we have been able to identify previously undetected functions of the phosphatase during meiosis, particularly those affecting meiotic recombination (Figure 8). Strikingly, we found the unprecedented requirement for Cdc14 to promote JM resolution at the transition between prophase I and the first meiotic division, suggesting that Cdc14 activity is not strictly ligated to its full release from the nucleolus.

### 3.1. Novel Insights into Cdc14 Meiotic Functions Using Different cdc14 Alleles

In the present work, we describe an allele of *CDC14* (*cdc14-HA*) that displays no obvious defects during unchallenged mitotic divisions, but it is strongly deficient in sporulation. Protein levels of the phosphatase are reduced in both mitotic and meiotic *cdc14-HA* cells, but this reduction is more dramatic during meiosis. However, the amount of the phosphatase in *cdc14-HA* meiotic cells is sufficient to complete meiotic DNA replication and both nuclear divisions with rather normal kinetics. Nonetheless, the *cdc14-HA* mutant fails to form spores once cells completed both nuclear divisions. It is possible that the impaired structural integrity of the SPBs observed in *cdc14-HA* can contribute to this phenotype. Similar defects in sporulation have been also described in the absence of components of the MEN pathway [62,92]. Importantly, correct SPB re-duplication and separation occur in a high proportion of *cdc14-HA* cells, but not in the more restrictive *cdc14-md* mutant that contains negligible levels of Cdc14 during meiosis. Furthermore, many *cdc14-HA* cells are able to assemble functional tetrapolar spindles at metaphase/anaphase II, confirming that problems arising in the *cdc14-HA* mutant somewhat differ from those occurring in *cdc14-md* as well as in the widely employed, *cdc14-1*, *cdc14-3* or other FEAR mutants [59,67,69]. This observation suggests that the small amount of the phosphatase still present in *cdc14-HA* is sufficient to carry out those roles required for meiotic SPB dynamics, but not to maintain SPB integrity after divisions. The *cdc14-HA* mutant also displays a short delay in cohesin removal when entering anaphase I, sharing this particular meiotic phenotype with *spo12*, *slk19* and *cdc14-1* mutants [59]. It is possible that delayed Rec8 removal from chromosomes could contribute to the presence of DNA bridges in late anaphase I. However, the proportion of *cdc14-HA* cells presenting anaphase I bridges is consistently higher than that of cells displaying persistent Rec8 signals during anaphase I; thus, it is unlikely that inefficient Rec8 removal is the only source of most DNA bridges, but instead a contributing factor. The fact that the formation of chromatin bridges in *cdc14-HA* and *cdc14-md* is substantially alleviated in the absence of meiotic DSBs strongly suggests that they may arise from chromosome entanglements, resulting from unresolved recombination intermediates, and raises the possibility of a direct role for Cdc14 in the regulation of meiotic recombination that has not been previously explored. Here, we demonstrate that, in some circumstances, Cdc14 is critically required for the proper resolution of meiotic JMs through the regulation of the Yen1 nuclease.

### 3.2. Multistage CDC14-Dependent Processing of Recombination Intermediates

Meiotic cells are endowed with a battery of enzymatic activities, including dissolvases and resolvases, that are tightly regulated to ensure the efficient and timely processing of recombination intermediates [17]. The processing of most meiotic JMs occurs at the end of pachytene [70], far earlier than the reported first release of Cdc14 at anaphase I. Nevertheless, this is not the first time that the requirement of Cdc14 has been linked to cell cycle stages preceding its bulk release from the nucleolus at anaphase. For example, DNA damage caused in vegetative cells triggers the transitory release of Cdc14 from the nucleolus to the nucleoplasm, allowing the phosphatase to act on components of the SPBs to stabilize them at metaphase [49]. Cdc14 is involved in the completion of late-replicating regions in the rDNA, and other parts of the genome [48]. Although the *cdc14-HA* mutant does not show a discernible mitotic phenotype under unchallenged conditions, it is formally possible that *cdc14-HA* cells, which contain reduced protein levels, may accumulate aberrant DNA structures during preceding mitoses prior to meiosis entry. However, this is unlikely the case for the *cdc14-md* mutant, whose mitotic protein levels are very similar to those of the wild type. Therefore, we favor a role for Cdc14 in the direct regulation of substrates required for correct processing of meiotic JMs (Figure 8). Interestingly, the conserved SSE, Yen1^GEN1^ is a critical substrate of Cdc14 during budding yeast mitosis, and it exhibits a phosphorylation-regulated nucleocytoplasmic shuttling behavior [93]. CDK-dependent phosphorylation restricts Yen1 from entering the nucleus and becoming active, whereas the reversal of that phosphorylation by Cdc14 allows Yen1 to enter the nucleus and resolve entangled DNA structures [23,24,86]. Nonetheless, Yen1 nuclear import and activation appears to be concomitant with bulk Cdc14 release from the nucleolus during anaphase. This is why a role for Yen1 in safeguarding chromosome segregation has been proposed, especially, during the second meiotic division [9,10,24]. Our results support this conclusion, since high frequencies of aneuploidy and mis-segregation events are detected during meiosis II in the *cdc14-HA* meiotic mutant when the other repair pathways are intact. This is consistent with the requirement for Cdc14 to activate Yen1. Puzzlingly, we also observe the Yen1-dependent processing of JMs in cells arrested during MI by Cdc20 depletion. This effect is manifested more prominently when the Mus81-Mms4 endonuclease and the Sgs1 helicase are missing. Such resolvase activity is not observed in *cdc14* mutants, suggesting that Cdc14 can activate Yen1 in cells that are about to initiate anaphase I. Taking into account that the massive Yen1 nuclear transport takes place during the ensuing anaphase [10], our results open the question of the origin of the Yen1 population acting in *CDC20*-depleted cells, which are stably arrested at the metaphase I to anaphase I transition in meiotic cells [74,85,92,94]. We speculate about other possible scenarios, such as the existence of a different fraction of Yen1 arising from a different subcellular compartment to perform this earlier role. In support of this possibility, it has been recently postulated that a subpopulation of Yen1 localizes at the nucleolus in mitotic cells [95]. Whether the nuclear Yen1 observed in MI is sourced from the cytoplasm or from other sub-nuclear compartment will be a matter of future studies.

### 3.3. Alternative Repair Pathways to Process JMs during Meiosis I

An intriguing possibility is that different SSEs might be required for processing different types of JMs that may not be necessarily good substrates for the canonical MutLγ complex. There is a risk that spindle forces originated during the *cdc20-md* arrest could generate tension at a number of still unprocessed JMs eventually forcing them towards conformations that cannot be easily processed by prophase-specific type of resolution/dissolution enzymes, for example affecting branch migration kinetics or directionality [18,96]. In such perilous scenario, Cdc14 could be an ideal regulator, together with Cdc5, to promote the switch between different endonucleases, by activating some while inhibiting others, once spindle forces start becoming dominant under circumstances where the SC no longer exists to hold homologs in close association [97,98,99].

In budding yeast meiosis, the timing of JM resolution and CO formation is coordinated with cell-cycle progression through the *NDT80*-dependent expression of the Polo-like kinase Cdc5 [9,19,20]. Thus, the requirement of Cdc5 for the resolution of dHJ at the pachytene to MI transition, upon Ndt80 activation, might also involve its ability to regulate the interaction of Cdc14 with Cif1/Net1 in the nucleolus. In opposition to mitotic cells, the action of Cdc5 could temporally counteract the negative regulatory effect of PP2A^Cdc55^ allowing some Cdc14 molecules to escape from its captor [66,67]. It is tempting to speculate that Cdc5 might also play a relevant role during meiosis in promoting an early, metastable, partial release of a Cdc14 population at the transition from the pachytene stage to metaphase I in order to modulate the activity of a number of safeguarding enzymes required for correct chromosome segregation [100]. Understanding the regulatory pathway/s which control Cdc14 activity prior to its full release during anaphase will be important to decipher its full contribution to the regulation of meiotic recombination.

In recent years, human orthologs of Cdc14 phosphatase have received increased attention due to their involvement in key processes such as DDR, DNA repair and cell cycle control. Furthermore, recent findings point to recessive variants of the phosphatase to be directly responsible for human diseases, such as deafness and male infertility [101]. Thus, in order to comprehend the underlying factors that trigger those conditions, a deeper understanding of the genetic and molecular mechanisms that are responsible for the countless functionalities of the Cdc14 phosphatase during gametogenesis and HR repair will be required.

## 4. Materials and Methods

### 4.1. Yeast Strains and Plasmids

All strains were SK1, as detailed in Appendix A. Standard yeast manipulation procedures and growth media were utilized. To introduce the 3HA tag at the C-terminal end of Cdc14, the CDC14ha3-pKan^R^ plasmid, containing the last ~500 bps of the *CDC14* gene with 3xHA inserted at the NotI restriction site and containing the *CLB1* terminator, was used. The plasmid was linearized using a unique restriction site located within the *CDC14* sequence and transformed into a SK1 haploid strain. Alternatively, *CDC14* was tagged using the PCR-based method described in [102] using the plasmid pFA6a-3HA-kanMX6. The phenotype of *cdc14-HA* strains obtained from both tagging methods was checked and the sporulation defect was identical. Transformants containing correct tag integration were identified and tested by Western blot for the presence of the tag. Southern blot analysis and/or PCR was performed to confirm the integration at the endogenous locus. Multi-copy plasmids used in Appendix A were originally described in [103].

### 4.2. Synchronous Meiotic Time Courses

Induction of synchronous meiosis was carried out according to the established protocols for standard assays [104]. All pre-growth and meiotic time courses were carried out at 30 °C unless specified otherwise. For *cdc14-1* meiosis, the culture was kept at 23 °C and shifted to 30 °C 2 h after transferring into sporulation medium (SPM). Aliquots were removed at the specified times and subjected to various analyses.

### 4.3. DNA Manipulation, Extraction and Southern Blot Detection

Standard DNA extraction was performed as in [105]. For studies at the *HIS4LEU2* recombination hotspot, the protocol described in [106] was followed. For 2D gel agarose electrophoresis, cell cultures were photo-crosslinked with trioxalen (Merck) using long-wave UV light before DNA extraction, in order to visualize recombination intermediates by standard Southern blotting techniques at the *HIS4LEU2* hotspot [106].

### 4.4. Time-Lapse Imaging, Immunofluorescence, Microscopy, and Image Analysis

Time-lapse experiments were performed as in [107], with small variations. In brief, 1 mL aliquots from synchronous meiotic cultures were taken at specific times and diluted 1:9 in fresh SPM (kept at 30 °C). Then, 300 µL of diluted cells were placed in suitable multi-well slides (80821 uncoated, ibidi, Gräfelfin, Germany). Slides were placed in a temperature-controlled incubation chamber from a Multidimensional Microscopy System Leica AF6000 LX. Images were taken at multiple positions and channels every 5, 10 or 15 min, depending on the experiment. Image acquisition was carried out using a CCD Hamamatsu 9100-02 camera. The system was controlled by the proprietary software, Leica Application Suite X (Wetzlar, Germany), v.3.7.4.23463. For preparations of fixed cells and immunofluorescence, aliquots were fixed and prepared as described in [10,105], respectively. Chromosomal DNA was stained with 1 µg/mL 4,6-diamino-2-phenylimide (DAPI). Images were recorded and analyzed using a Deltavision (DV3) workstation from Applied Precision Inc. (Mississauga, ON, Canada) with a Photometrics CoolSnap HQ (10-20MHz) air cooled CCD camera and controlled by SoftWorx v.6.5.2 image acquisition and deconvolution software. 

### 4.5. Protein Extraction, Western Blot Analysis and Antibodies

Whole-cell extracts were prepared from cell suspensions in 20% trichloroacetic acid by agitation with glass beads. Precipitated proteins were solubilized in SDS-PAGE sample buffer, and appropriate dilutions were analyzed by SDS-PAGE and Western blotting [108]. Antibodies used for Western blotting were mouse monoclonal anti-MYC (1:1000, Abcam, Cambridge, UK), mouse monoclonal anti-HA (1:1000) from S. Ley (NIMR), goat polyclonal anti-Cdc14 (yE-17; 1:1000; Santa Cruz Biotechnology, Dallas, TX, USA), goat polyclonal anti-Cdc5 (1:1000; Santa Cruz Biotechnology), mouse monoclonal anti-Pgk1 (1:5000; Invitrogen, Waltham, MA, USA), goat anti-mouse IgG conjugated to horseradish peroxidase (1:10,000; Sigma-Aldrich, Burlington, MA, USA), and chicken anti-rabbit IgG conjugated to horseradish peroxidase (1:10,000; Sigma-Aldrich, Burlington, MA, USA).

### 4.6. Nuclease Assays

For nuclease assays, myc9-tagged Yen1 and Yen1^ON^ were immuno-affinity purified from yeast using anti-Myc agarose beads (9E10) and washed extensively. The beads (approx. volume 10 µL) were then mixed with 10 µL cleavage buffer (50 mM Tris-HCl pH 7.5, 3 mM MgCl_2_) and 15 ng of 5′-Cy3-end-labeled synthetic Holliday junction X0 DNA. After 1 h incubation at 37 °C with gentle rotation, reactions were stopped by the addition of 2.5 µL of 10 mg/mL proteinase K and 2% SDS, followed by incubation for 30 min at 37 °C. Loading buffer (3 µL) was then added and fluorescently labelled products were separated by 10% native PAGE and analyzed using a Typhoon scanner and ImageQuant software version 5.0. Resolution activity was calculated by determining the fraction of nicked duplex DNA product relative to the sum of the intact substrate and resolution product. The protein input was estimated by Western blot.

### 4.7. Data Analysis and Biostatistics

Data were compiled and analyzed using Excel, LibreOffice Calc, and SPSS Statistical Data Editor. For multiple comparisons, analysis of variance (one-way ANOVA) was performed. For pairwise comparisons, two-tailed unpaired *t*-tests were used using IBM SPSS Statistics version 27 and SigmaPlot V13.

## Figures and Tables

**Figure 1 ijms-22-09811-f001:**
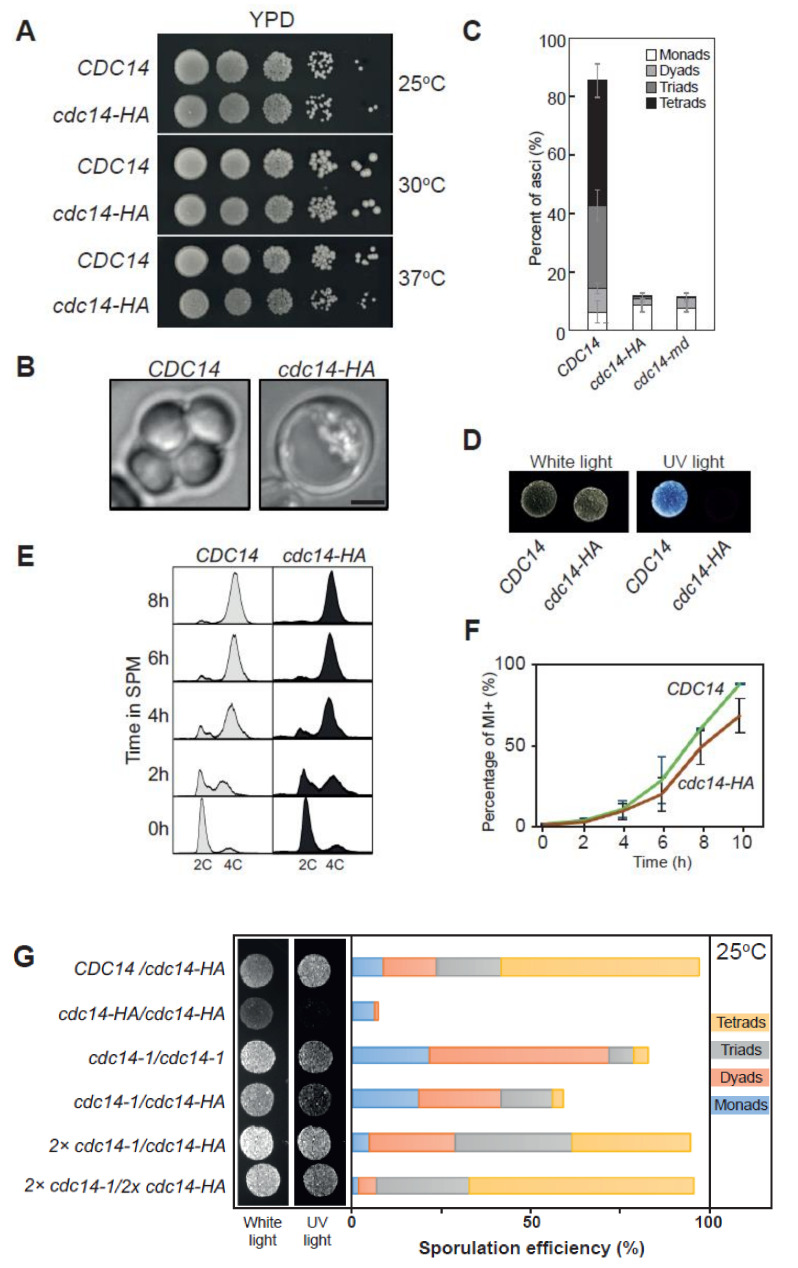
*cdc14-HA* behaves as a specific sporulation-deficient separation-of-function allele of *CDC14*. (**A**) The *cdc14-HA* allele does not perturb normal growth conditions when cultivated at 25 °C, 30 °C and 37 °C in rich media (YPD). (**B**) Homozygous *cdc14-HA* diploids do not form asci containing spores under standard conditions for sporulation. Scale bar represents 1 µm. (**C**) Percentage of sporulation in *CDC14* (JCY840), *cdc14-HA* (JCY844) or *cdc14-md* (JCY2376). Error bars represent the standard deviation of the mean (SEM) calculated from at least three independent experiments. A minimun of 300 cells per strain were counted. (**D**) *cdc14-HA* diploid cells divide mitotically but lack di-tyrosine autofluorescence induced by UV light after several days of incubation in SPM media at 30 °C. (**E**) FACS analysis of DNA content shows that *cdc14-HA* cells complete meiotic DNA replication with identical kinetics as *CDC14* diploid cells. (**F**) *cdc14-HA* cells undergo meiotic nuclear divisions at subtly slower kinetics, and reduced frequencies, than *CDC14* cells. Error bars represent the SEM over the mean values plotted. (**G**) Di-tyrosine autofluorescence of different mutant combinations as well as the control strains grown and sporulated on plates at 25 °C. *cdc14-1* homozygous diploids (JCY2353) sporulate at high efficiency under semi-permissive temperature forming preferentially tetrads whereas *cdc14-HA* homozygous diploids (JCY844) do not sporulate. Combinations, and variable copy number, of the mutant genes can rescue the sporulation defect to different degrees (JCY2365/JCY2354/JCY2356).

**Figure 2 ijms-22-09811-f002:**
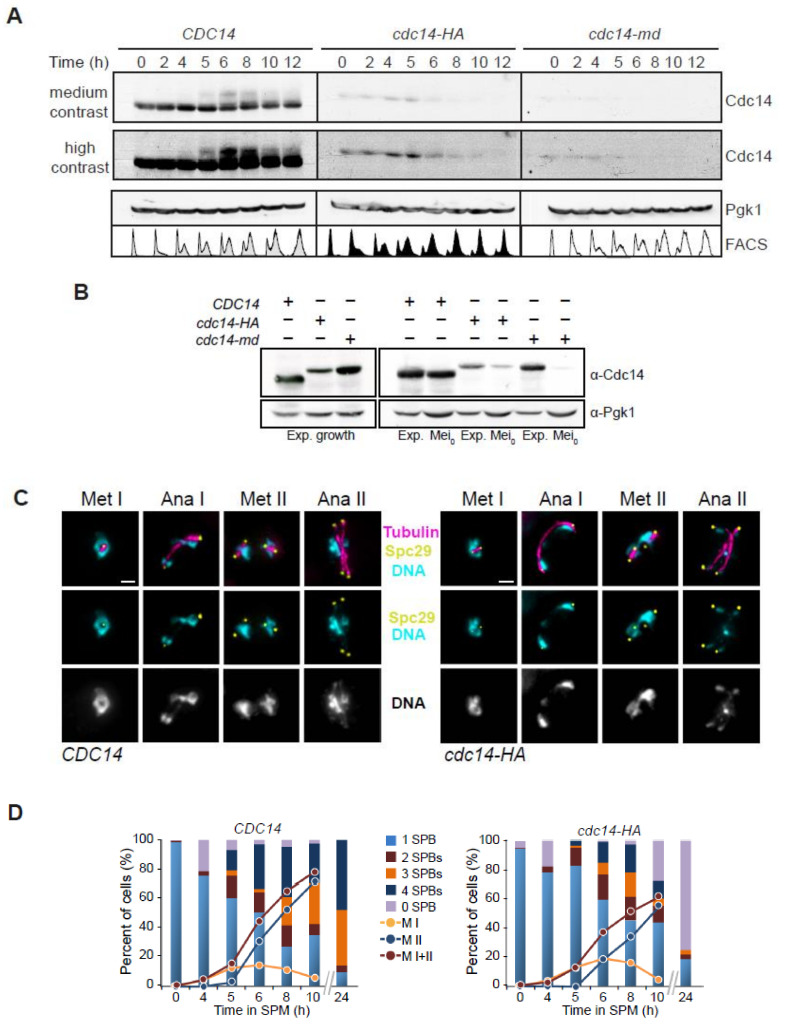
Diminished Cdc14 protein levels in *cdc14-HA* meiotic cells compromise SPB integrity following meiotic divisions. (**A**) Dynamics of Cdc14 protein levels throughout meiosis in synchronous time courses for *CDC14* (JCY2232), *cdc14-HA* (JCY2231), and *cdc14-md* (JCY2389). Higher contrast was applied to visualize residual protein levels for all three strains. Immunodetection of Pgk1 was used as a loading control. FACS histograms for each time-point are depicted at the bottom to show the degree of synchrony reached in each culture. (**B**) Variability of protein levels in mitotic versus meiotic cells in different *CDC14* alleles. Exponentially growing cells show similar Cdc14 levels, excluding *cdc14-HA*, which has lower overall protein levels (left panel). Direct comparison between the three alleles *CDC14* (JCY2232); *cdc14-HA* (JCY2231) and *cdc14-md* (JCY2389) in exponentially growing mitotic cells (Exp.) and meiotic cultures immediately after they were transferred to SPM (Mei_0_). The loading control for each lane was determined using Pgk1. (**C**) Visualization of Spc29-CFP at SPBs, tubulin, and DNA, in wild-type (JCY892) and *cdc14-HA* (JCY893) cells fixed at different stages of meiosis at 30 °C. Scale bar represents 1 µm. (**D**) *cdc14-HA* cells develop meiosis I and II spindles with near wild-type kinetics. Lack of spore formation in *cdc14-HA* meiotic cells parallels the loss of SPB’s structural integrity.

**Figure 3 ijms-22-09811-f003:**
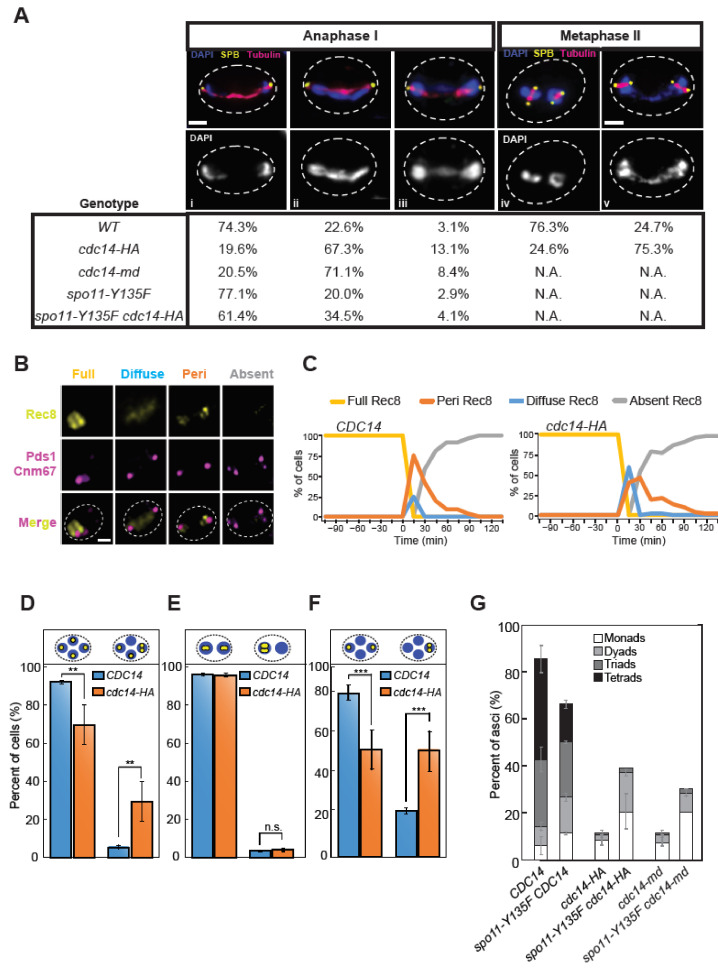
Mis-segregation of sister chromatids at MII in meiosis-deficient *cdc14* mutant cells. (**A**(**i**–**iii**)) Frequency of cells presenting connected DAPI masses at anaphase I in wild type (JCY902), *cdc14-HA* (JCY904), *cdc14-md* (PRY132), *spo11-Y135F* (PRY53) and *spo11-Y135F cdc14-HA* (PRY55). (**A**(**iv**,**v**)) Frequency of cells presenting connected nuclei at metaphase II. Meiotic-deficient *cdc14-HA* mutants (GGY54) present higher frequencies of unresolved nuclear divisions at late anaphase I and at metaphase II than wild type (GGY53). (**B**) Representative images of distinct morphological patterns for Rec8-GFP in meiosis. Full: bright nuclear localization. Diffuse: faint staining within the stretched nucleus. Peri: Rec8-GFP only visible at peri-centromeric locations. Absent: no Rec8-GFP signal. Pds1-tdTomato and CNM67-tdTomato were used to follow Pds1 accumulation/degradation and SPB number/location. (**C**) Temporal distribution of the distinct morphological categories of Rec8-GFP shown in (**B**) for *CDC14* (JCY2406; *n* = 57 cells) and *cdc14-HA* (JCY2404; *n* = 52 cells). (**D**) Unequal distribution of homozygous chromosome II-linked GFP markers at lys2 locus in *cdc14-HA* mutant cells (JCY2330) denotes increased chromosome missegregation in comparison with wild-type cells (JCY2331). (**E**) Similar distribution of homozygous *CENIV*-linked GFP markers at *trp1* locus in meiosis-deficient *cdc14-HA* mutant cells (JCY2286) in comparison with wild-type cells (JCY2284) denotes correct homolog disjunction in anaphase I. (**F**) Unequal distribution of heterozygous *CENIV-GFP* marker in meiosis-deficient *cdc14-HA* tetranucleated mutant cells (JCY2327) denotes increased sister chromatid missegregation in comparison with wild-type cells (JCY2326). Statistical significance of differences was calculated by two-tailed *t*-test, assuming unequal variances (* *p* < 0.05; ** *p* < 0.01; *** *p* < 0.001; **** *p* < 0.0001; n.s.: not significant). (**G**) Sporulation defects in *cdc14-HA* (JCY844) or *cdc14-md* (JCY2376) can be partly alleviated by eliminating recombination (JCY2270/JCY2280/PRY151). *CDC14* (JCY840). Error bars represent SEM over the calculated mean value from three independent experiments. A minimum of 300 cells per strain were counted.

**Figure 4 ijms-22-09811-f004:**
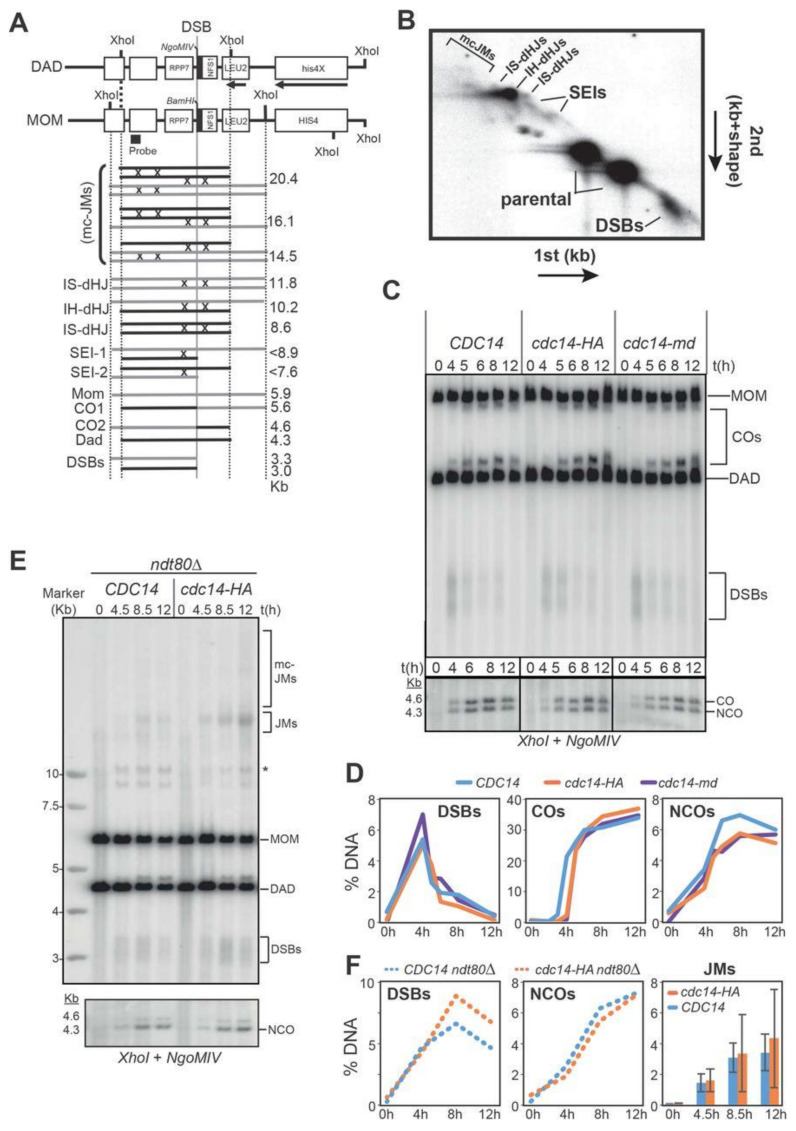
Analysis of meiotic recombination in meiotic-deficient *cdc14* mutants. (**A**) Schematic illustration of recombination analysis and intermediates at the *HIS4LEU2* hotspot (see Results for more details). (**B**) Schematic representation of DNA species commonly observed after analyzing the *HIS4LEU2* hotspot by 2D-gel electrophoresis. ((**C**); **top**) Representative 1D-gel Southern blot for analysis of DSBs and COs at the HIS4LEU2 hotspot using XhoI as the restriction site. Wild-type strain (JCY2232), *cdc14-HA* (JCY2231), and *cdc14-md* (JCY2389). ((**C**); **bottom**) Analysis of COs and NCOs at the *HIS4LEU2* hotspot using *XhoI* and *NgoMVI* in the same strains as in (**C**). (**D**) Quantification of DSBs, COs and NCOs from the Southern blots shown in (**C**). (**E**) Representative 1D-gel Southern blot image for analysis of DSBs, COs (top) and NCOs (bottom) at the *HIS4LEU2* hotspot in *CDC14 ndt80Δ* strain (JCY2390) and *cdc14-HA ndt80Δ* (JCY2385) strains. Asterisk indicates meiosis-specific non-characterized recombination products. (**F**) Quantification of DSBs, NCOs and JMs from the Southern blots shown in (**E**). Error bars represent SEM over the calculated mean value.

**Figure 5 ijms-22-09811-f005:**
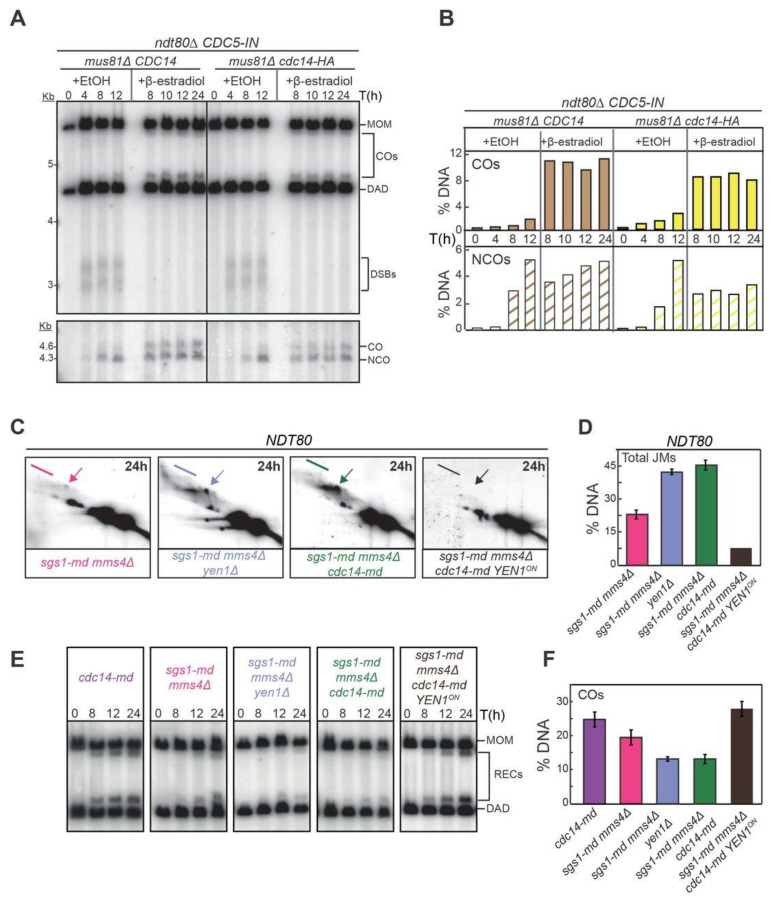
Analysis of recombination intermediates in *cdc14* mutants shows early and late accumulation of JMs. (**A**) Representative Southern blots depicting 1D-gels at the *HIS4LEU2* hotspot following induction of *CDC5* in prophase-arrested *ndt80Δ mus81Δ cdc14-HA* cells (JCY2440) leads to inefficient CO and NCO formation compared to *ndt80Δ mus81Δ* (JCY2442). (**B**) Quantification of COs and NCOs from Southern blots shown in (**A**). (**C**) Representative Southern blots depicting 2D-gels at the *HIS4LEU2* hotspot in *sgs1-md mms4Δ* (JCY2444), *sgs1-md mms4Δ cdc14-md* (JCY2446), *sgs1-md mms4Δ yen1Δ* (JCY2448) and *sgs1-md mms4Δ cdc14-md YEN1^ON^* (JCY2529) 24 h into meiosis. Arrows mark IH-dHJs and lines mark mc-JMs (see Figure 4A,B). (**D**) Quantification of total JMs in the strains shown in (**C**) from at least two independent gels. Error bars represent SEM over the calculated mean value. (**E**) Representative 1D-gel Southern blot images for analysis of crossovers at the *HIS4LEU2* hotspot for all strains shown in (**C**) and for *cdc14-md* (JCY2389). (**F**) Quantification of COs from at least three different image acquisitions such as that depicted in (**E**). Error bars represent SEM over the calculated mean value.

**Figure 6 ijms-22-09811-f006:**
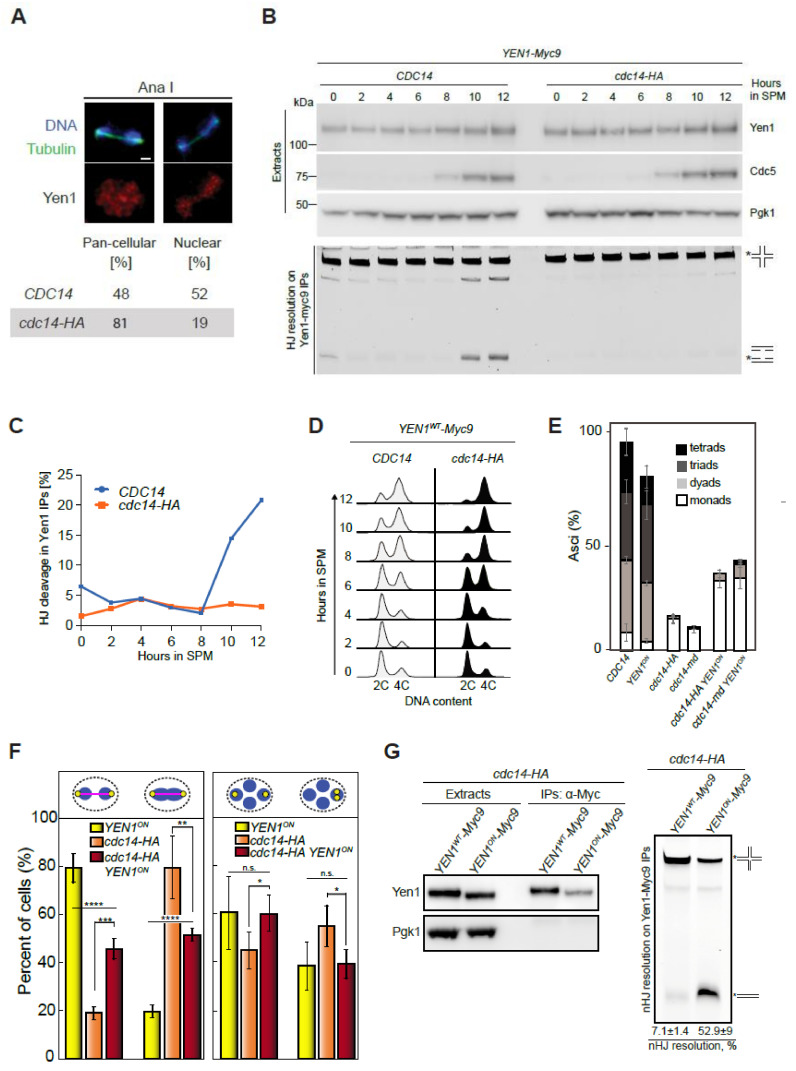
The absence of Cdc14 prevents the activation of Yen1 during meiotic divisions. (**A**) Quantification of distinct localization patterns of Yen1 at meiosis I. Scale bar represents 1 µm. (**B**) Analysis of expression levels and nuclease activity of Yen1 in *CDC14* (YML7692) and *cdc14-HA* (YML7693) meiotic cells. Soluble extracts were prepared from *YEN1-Myc9* strains at 2 h intervals after transfer into sporulation medium (SPM). Following anti-Myc immunoaffinity purification (IP), the IPs were analyzed by Western blotting and tested for nuclease activity using Cy3-labeled Holliday junction DNA as a substrate. Upper panel: Western blots of the cell extracts, with detection of Yen1-myc9, Cdc5, and Pgk1 (loading control). Lower panel: HJ resolution assay. The experiment shown is representative of two independent experiments. (**C**) Percentage of HJ cleavage in Yen1 IPs during meiosis in wild type and *cdc14-HA*. (**D**) Evolution of DNA content during meiosis from strains in (**B**). (**E**) Unrestrained resolution of recombination intermediates by Yen1^ON^ improves sporulation in *cdc14-HA* (JCY2164) and *cdc14-md* (PRY182) cells. Frequency of asci containing one, two, three and four spores in the strains of the indicated genotypes. Error bars represent SEM over the calculated mean value from three independent experiments. A minimum of 300 cells per strain were counted. (**F**) Frequency of cells presenting connected DAPI masses at anaphase I (**left panel**) or sister chromatid mis-segregation (**right panel**) in *YEN1^ON^* (PRY123/PRY99), *cdc14-HA* (JCY2327/PRY55) and *cdc14-HA YEN1^ON^* (PRY121/PRY108). Statistical significance of differences was calculated by two-tailed *t*-test, assuming unequal variances (* *p* < 0.05; ** *p* < 0.01; *** *p* < 0.001; **** *p* < 0.0001; n.s.: not significant). (**G**) Poor HJ resolution in *cdc14-HA* (YML7693) is efficiently restored by the presence of Yen1^ON^ (JCY2421). Western blot analysis of Yen1/Yen1^ON^ immunoprecipitates is shown on the left panels. HJ resolution assay is shown on the right panel. Quantification of resolution efficiency is displayed at the bottom. Resolution data arise from two independent experiments. Asterisks in (**B**) and (**G**) indicate labelled DNA strands.

**Figure 7 ijms-22-09811-f007:**
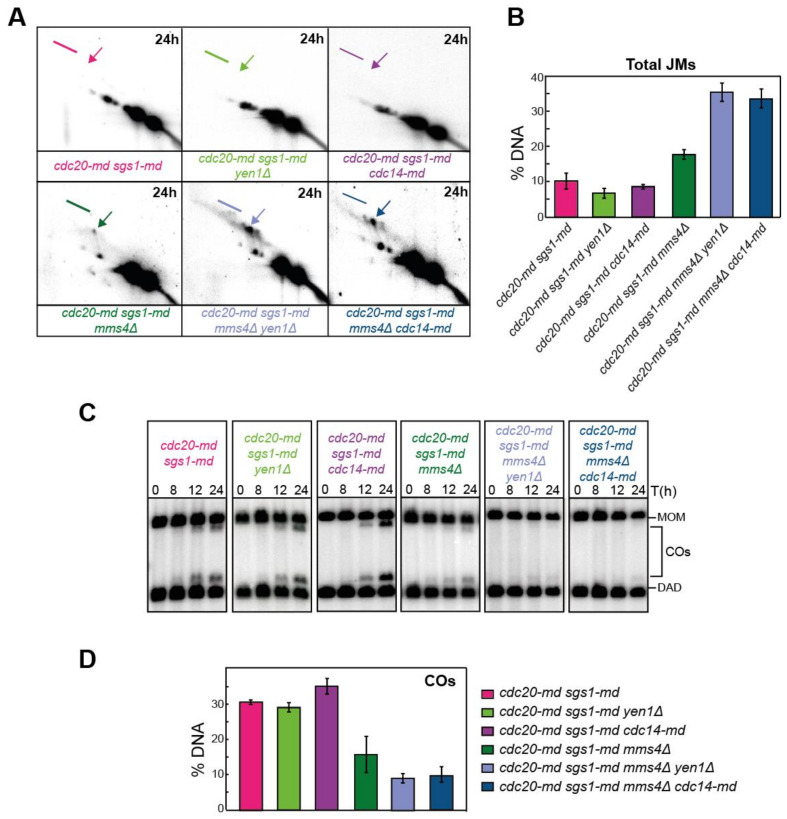
Analysis of recombination intermediates in the *cdc20-md* mutant reveals an early meiotic role for Cdc14 and Yen1. (**A**) Representative Southern blots depicting 2D-gels at the *HIS4LEU2* hotspot in *CDC14 cdc20-md sgs1-md* (JCY2480), *cdc20-md sgs1-md yen1* (JCY2469), *cdc20-md sgs1-md cdc14-md* (JCY2508), *cdc20-md sgs1-md mms4* (JCY2480), *cdc20-md sgs1-md mms4 yen1* (JCY2478) and *cdc20-md sgs1-md mms4 cdc14-md* (JCY2502). Arrows mark IH-dHJs and lines mark mc-JMs (see Figure 4A,B). (**B**) Quantification of total JMs in the strains shown in (**A**) from several independent gels. Error bars display SEM over the mean values plotted. (**C**) Representative 1D-gel Southern blot images for analysis of crossovers at the *HIS4LEU2* hotspot for all strains shown in (**A**). (**D**) Quantification of COs from at least three different image acquisitions, such as that depicted in (**C**). Error bars represent SEM over the calculated mean value.

**Figure 8 ijms-22-09811-f008:**
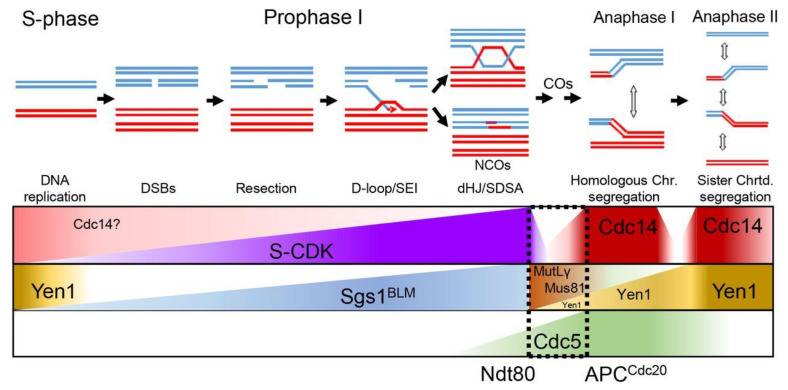
Model for *CDC14*-dependent resolution of recombination intermediates via multiple mechanisms. Contribution of Cdc14 to the correct disjunction of recombining chromosomes during meiosis. Early in prophase I, chromosomes initiate homologous recombination. 3′-resected ssDNA overhangs invade an intact template with the help of recombinases. The displacement of the intact strand from the template allows for the formation of D-loops, which can be stabilized, allowing for DNA synthesis. Next, stabilized branched DNA molecules might be disrupted by the action of the anti-recombinases. Reannealing of the extended 3′-ssDNA overhang with the resected complementary strand, followed by further DNA synthesis will lead to the repair and ligation of the broken DNA duplex, giving rise to NCOs via SDSA. ZMM stabilization of JMs is followed by the resolution of dHJs through the MutLγ, class I, dedicated CO pathway. Ndt80-dependent expression, and activation, of Cdc5 triggers MutLγ resolvase activity. Unresolved linkages between bivalents that persist until metaphase/anaphase I are mostly resolved by the action of the SSEs, Mus81-Mms4. Slx1-Slx4, Top3-Rmi1 as well as Cdc14/Yen1 also contribute to the correct resolution of chromosomal entanglements between homologs during MI. Residual chromatid intertwining between sister chromatids during the second meiotic division will be removed by the action of Cdc14/Yen1. Gradual implementation of Yen1 activity during both divisions by Cdc14 will transfer Yen1 inactive population to its active/nuclear-enriched form.

## Data Availability

Not applicable.

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
