# Peer review of "The Cdc14 Phosphatase Controls Resolution of Recombination Intermediates and Crossover Formation during Meiosis"

_ijms, 2021, doi:10.3390/ijms22189811_

Round 1
Reviewer 1 Report
The manuscript by Alonso-Rsmos et al. employed multiple approaches to reveal an important role for Cdc14 phosphatase in regulation of resolving recombination intermediates and crossover formation during meiosis. They generated Cdc14 defective yeast and found that deficiency of Cdc14 caused abnormal segregation of sister chromatids during meiosis and also impaired meiotic recombination. They then found that Cdc14 regulates subcellular localization of resolvase Yen1 during anaphase I, which is important for resolution of Holliday Junction. Further, the nuclease activity of Yen1 is also stimulated by Cdc14 when Cdc5 is highly active. Finally, the authors reported that Cooperation of Cdc14 and Yen1 ensures correct segregation of sister chromatid during the second meiotic division.
Overall, this manuscript reports a novel and important role of Cdc14 in regulation of meiotic recombination in yeast cells. The related experiments were carefully designed and performed, and the resulting data are properly explained to support their hypothesis. The while manuscript is well prepared to I would like to endorse publication of this manuscript in International Journal of Molecular Science.
Some minor concerns are listed below.
- Line 54, Mus81/Mms4 should be Mus81-Mms4. Same, Slx1/Slx4 should be Slx1–Slx4.
- Fig 2A, “hours” can be added beside the numbers.
- Some panels in figures do not appear in main text as orders, like Fig. 3G in Line 215;Fig. 4B in Line 413.
Author Response
Response to Reviewer 1 Comments
POINT 1: Line 54, Mus81/Mms4 should be Mus81-Mms4. Same, Slx1/Slx4 should be Slx1–Slx4.
RESPONSE 1: Thank you to the reviewer for pointing out this missannotation when listing protein complexes. We have now changed them using the suggested nomenclature. In addition to those listed by the reviewers, we found few others in the text, that we have now replaced with the right wording.
POINT 2: Fig 2A, “hours” can be added beside the numbers.
RESPONSE 2: Thank you for perceiving this omission when labelling the figure. Hours (h) have now been added to the new figure 2A.
POINT 3: Some panels in figures do not appear in main text as orders, like Fig. 3G in Line 215;Fig. 4B in Line 413.
RESPONSE 3: Many thanks for pointing out this issue which creates some confusion in the order of figures and results. This was also a matter of concern for other reviewer. We have now labelled all panels in figures to match their order they first are found in the text.
To sort out the figure 3G panel, we have now move the relevant cited data to a new panel to Figure 1, being now figure 1C in the new figure 1, since it was redundant with the earlier graphs depicted in the former figure 1A. Al further figure 1 panels have been updated and renamed in the text to match those of the new Figure 1. The new Figure 3G panel now has been reorganized to match the data cited in the text.
Reviewer 2 Report
Alonso-Ramos and Alvarez-Melo et al. make use of two clever mutants of the essential Cdc14, and were uniquely poised to investigate the role of Cdc14 in meiotic recombination. Using a sequence of well thought out and rationally laid out extensive array of experiments, they manage to convincingly unravel that and potentially more.
This work is thorough and suitable for publication in IJMS, pending minor revisions to the manuscript. General comments:
- Writing style:
This is important piece of work and so, it is our collective responsibility to ensure that the readers can access it as much as the authors intend them to. In its current form, the manuscript suffers from being longwinded in its writing. Many sections can highly benefit from brevity in writing. For instance, the introduction feels like a chapter – while it is a great educational resource for the reader outside the field, it necessarily takes the attention away from the crux of the study – novel roles of Cdc14 in meiosis. The discussion section could use less of internal references to figures in the manuscript, as the results section serves that purpose already.
- Organization of the results section:
The results section is rather difficult to navigate due to jumbled call outs to figures, especially in the initial parts. To the extent possible, it will become much easier to read if some figure panels are rearranged to flow better; and if the subtopics of the results section are one per figure.
- Figure statistics:
Many figure panels are missing error bars and/or statistical significance notations. The figure legends could benefit from the mentioning of how many cells/replicates were done, etc. Despite the multiple lines of inquiry giving rise to overall convincing evidence, this is rather important, because the confidence with which to interpret a piece of data will change with the technical rigor.
Specific comments in the order of appearance in the manuscript:
Abstract:
- Line 20: Clarify that RNA repair means the resolution of crossovers, in this context of meiosis.
- A mention about the conservation of Cdc14/ Cdc14-like mechanism across eukaryotes will motivate a broader readership to take interest in the manuscript. This is mentioned in the introduction, but doing so in the abstract additionally, will help.
Results:
- Line 168: Even if it has been characterized before in another study, it is very important to mention here how the addition of a simple HA-tag eliminates activity/stability of Cdc14 in meiosis specifically. Is this the full-length protein, otherwise? This was stunning, but also an elephant in the room for me, and made it difficult to read on.
- Fig S1C: mention how you delineated the boundary of the meiotic cell.
- Line 362: change “Cdc14 activity” to “Cdc14 levels” as a more accurate interpretation of the data.
- Line 466 and Fig. 6A: Is Yen1 as important as written, given that even in the WT CDC14 background, only 52% of cells have Yen1 in the nucleus in Anaphase I? Clarify. Is this due to technical difficulty of accurately delineating the cells in the right phase? If so, would you see 100% cells with nuclear Yen1 in synchronized culture of some sort?
- Line 476: how is the robust HJ processing in that two-hour window only due to Yen1? This interpretation seems too generous. For instance, Cdc5 goes up in that time window, but Yen1 levels stay steady. The HJ processing could be some other effect of Cdc5 as well. So, the current interpretation needs to be toned down, if not, please clarify better.
Author Response
Response to Reviewer 2 Comments
POINT 1: This is important piece of work and so, it is our collective responsibility to ensure that the readers can access it as much as the authors intend them to. In its current form, the manuscript suffers from being longwinded in its writing. Many sections can highly benefit from brevity in writing. For instance, the introduction feels like a chapter – while it is a great educational resource for the reader outside the field, it necessarily takes the attention away from the crux of the study – novel roles of Cdc14 in meiosis. The discussion section could use less of internal references to figures in the manuscript, as the results section serves that purpose already.
RESPONSE 1: We thank the reviewer for this important comment. We have focused the introduction to the scope of this study so to avoid distracting the readers with broader knowledge about the topics involved. We have done that while trying to keep some relevant background information for readers less familiarized with the field, by keeping the cited bibliography at the right places, when possible, so the more curious readers can find the omitted info in their original studies. We think we have reached a reasonable compromise between both alternatives.
The new text has now replaced several paragraphs (former lines 103 to 108 were eliminated, and the penultimate paragraph has been reorganized and lines 137-139 were replaced by a new sentence.
In the discussion, we have now removed figure citations from the results, except for figure 8, which displays our current working model.
POINT 2: The results section is rather difficult to navigate due to jumbled call outs to figures, especially in the initial parts. To the extent possible, it will become much easier to read if some figure panels are rearranged to flow better; and if the subtopics of the results section are one per figure.
RESPONSE 2: We thank this reviewer for bringing this point to our attention. We have attempted to improve the writing style in the introduction to make it more clear, and avoid unnecessary duplicities when calling figures in the text, especially when these figure panels alter their assigned numerical or alphabetical order in the text. A similar point was also made by other reviewer as some figure panels were called in the text before the order they were given in the figures. We have now fixed those commented by both reviewers, by creating a new panel “1C” in the new Figure 1. We have also included in the text an earlier call to Figure 4B, so now it also follows the numerical and alphabetical order in the text. In addition, we have removed redundant calls in supplementary figures, disregarding to calling twice at analogous results leading to same conclusion. Even though we put our best efforts to fix this issue raised by the reviewers, on the other hand, the high number of results that are packed in the supplementary data as part of the phenotype characterization of the cdc14-HA allele, which was not described before, makes extremely hard to eliminated all those multiple figure calls in the text, otherwise omitting important information for the readers.
Regarding reducing the number of subtopics, we think it is better to leave it in its current form, due to the fact that many figures display information that lead to different, or unrelated, conclusions, so we cannot summarize them in one single subtopic. The alternative is to increase the number of figures to match each relevant subtopic in the manuscript, but we think that eight figures is already a large number for a single manuscript, and reorganizing them could create more confusion for readers not familiar with the topic.
POINT 3: Many figure panels are missing error bars and/or statistical significance notations. The figure legends could benefit from the mentioning of how many cells/replicates were done, etc. Despite the multiple lines of inquiry giving rise to overall convincing evidence, this is rather important, because the confidence with which to interpret a piece of data will change with the technical rigor.
RESPONSE 3: Thank you for raising this important question. We have now added relevant statistics to graphs in the new Figure 1C and new Figures 6E and 6F, to convey with the editor’s request. Also we have added the description of the statistical analysis in the figure legends, including those that were not originally added in already existing analysis. Numbers of cells, repeats, etc. were included in figure legends or experimental procedures for most of the relevant results.
On the other hand, in order to avoid cluttering some graphs with excessive numbers of data points, error bars, p-values, etc.; some results, especially those that are displayed in figures to describe a trend, or a behaviour, and that later are corroborated (or disregarded) by more robust statistical analysis using the same or a different experimental assay, we prefer to leave them as they were in their original form, for clarity. For that reason, we prefer to keep some of the graphs in their original state (uncluttered state), following the same figure style trends that many experts in the recombination field typically use.
POINT 4: Line 20: Clarify that RNA repair means the resolution of crossovers, in this context of meiosis.
RESPONSE 4: We understand that the reviewer was meant to write DNA repair, and not RNA repair. In the case of DNA repair, we think that it is best not to limit the type of repair that takes place in meiosis to the resolution of crossovers, since other types of DNA repair pathways which participate, even under unchallenged wild-type condition, play an important part during the repair of meiotic DSBs, equally important to avoid mutations, translocations/deletions or all sort of forms of defects that lead to genomic instability, including aneuploidies. Due the complexity of the entire recombination process that comprise from the initiation of recombination until the formation of the final recombination products, we prefer to leave this statement in its current form so not to mislead the reader into believing that only a single repair pathway plays a relevant part during meiotic recombination.
POINT 5: A mention about the conservation of Cdc14/ Cdc14-like mechanism across eukaryotes will motivate a broader readership to take interest in the manuscript. This is mentioned in the introduction, but doing so in the abstract additionally, will help.
RESPONSE 5: We thank this reviewer for raising this important point. Although the likeness that a Cdc14-like mechanism might be conserved across eukaryotes remains to be demonstrated in other organisms outside S. cerevisiae, considering that Cdc14 is required for the reversal of CDK phosphorylation in most eukaryotes that present an ortholog of the phosphatase, it is likely that in these organisms several targets of Cdc14 are also important for accurate meiotic DNA repair. We have now adapted the abstract to include this possibility.
POINT 6: Line 168: Even if it has been characterized before in another study, it is very important to mention here how the addition of a simple HA-tag eliminates activity/stability of Cdc14 in meiosis specifically. Is this the full-length protein, otherwise? This was stunning, but also an elephant in the room for me, and made it difficult to read on.
RESPONSE 6: We apologize for this confusion, we have now attempted to clarify this part. This is the first study that describes a meiotic phenotype for this allele of CDC14. As such, we employed a big part of the initial result section to describe the meiotic phenotype for the cdc14-HA allele. We have now made several changes in the referred sentence to address some of the confusion which might have make unclear for the reviewer what was the origin of this allele. As an added note, this was not entirely surprising for us, since C/N-terminal addition of frequently used tags to full-length proteins often affects their functionality. What it was more unexpected was that in this particular case, those defects in the tagged protein appeared to occur more drastically in meiotic cells than in mitotic cells and perhaps, that is the reason why it went unnoticed in other mitotic studies where they used the same allele for a different purpose.
POINT 7: Fig S1C: mention how you delineated the boundary of the meiotic cell.
RESPONSE 7: The images depicted in figure S1C are fluorescent images to visualize the nuclear DNA as this was stained with DAPI. To confirm that those cells showing more than one nucleus was not due to having photographed a cluster of cells, we confirmed that they were individual cells by first looking at bright field microscopy. Due to some of those images were obtained using different microscopes, we omitted the bright field images since they were acquired using different methodologies (phase contrast or DIC/Nomarski), and might complicate the interpretation of this result from the supplementary section. We have now added a sentence to the supplementary figure legend S1C to clarify this matter.
POINT 8: Line 362: change “Cdc14 activity” to “Cdc14 levels” as a more accurate interpretation of the data.
RESPONSE 8: We thank the reviewer for perceiving this inaccuracy, we have now changed that to its correct form.
POINT 9: Line 466 and Fig. 6A: Is Yen1 as important as written, given that even in the WT CDC14 background, only 52% of cells have Yen1 in the nucleus in Anaphase I? Clarify. Is this due to technical difficulty of accurately delineating the cells in the right phase? If so, would you see 100% cells with nuclear Yen1 in synchronized culture of some sort?
RESPONSE 9: We appreciate the point raised by the reviewer in this particular situation. We have toned down the sentence and we have eliminated the word “markedly” to avoid such type of overstatements in the results section. Nonetheless, we still believe that the reduction of nuclear Yen1 during anaphase I in the cdc14-HA mutant is considerable, and what the wild type control does not fully reflect it using this type of assay because of the naturally lower synchrony of meiotic cells compared with mitotic cells entering anaphase in this organism. As example, mitotic cells, better synchronized, only displayed around 80% of nuclear Yen1 in anaphase [1]. Additionally, anaphase I progresses very rapidly in budding yeast, so it is hard to capture cells in such stage, as pointed out by the reviewer. In support to our quantified data, other proteins relevant for the resolution of meiotic recombination cannot even be detected by this methodology, thus we believe that our results are particularly valuable here.
POINT 10: Line 476: how is the robust HJ processing in that two-hour window only due to Yen1? This interpretation seems too generous. For instance, Cdc5 goes up in that time window, but Yen1 levels stay steady. The HJ processing could be some other effect of Cdc5 as well. So, the current interpretation needs to be toned down, if not, please clarify better.
RESPONSE 10: We do not fully agree with this concern, considering that the assay the reviewer is refereeing to was performed using purified Yen1 protein from the relevant control and mutant cultures, therefore it was very likely that Yen1 was the only nuclease that could cut so efficiently the in vitro substrate used in this type of assay. On the other hand, since Cdc5 protein was produced at similar kinetics and amounts in both cultures, its presence should affect to similar extent to both cultures, but not differentially. The only difference in such experiment was the presence or not of the cdc14-HA mutation when Yen1 was immunoprecipitated. As the reviewer pointed out, total levels of Yen1 in the extract was unaffected, but it is important to distinguish that Cdc14 modifies the phosphorylation status of Yen1 and not its total protein levels. Thus we believe that our initial interpretation is correct. We hope, we have now better clarified this point.
References
- Blanco, M. G.; Matos, J.; West, S. C., Dual control of Yen1 nuclease activity and cellular localization by Cdk and Cdc14 prevents genome instability. Mol Cell 2014, 54, (1), 94-106.